# Nurse Cultural Competence-cultural adaptation and validation of the Polish version of the Nurse Cultural Competence Scale and preliminary research results

Danuta Zarzycka[1], Agnieszka Chrzan-Rodak[2]*, Jadwiga Bąk[3], Barbara Niedorys-Karczmarczyk[3], Barbara Ślusarska[2]

1 Department of Paediatric Nursing, Faculty of Health Sciences, Medical University of Lublin, Lublin, Poland, 2 Department of Family Medicine and Environmental Nursing, Faculty of Health Sciences, Medical University of Lublin, Lublin, Poland, 3 Faculty of Health Sciences, Medical University of Lublin, Lublin, Poland

* agnieszkachrzan607@gmail.com

**Data Availability Statement:** All relevant data are within the paper and its Supporting Information files.

## Abstract

### Introduction

Measuring nurses' cultural competence is an important aspect in monitoring the acceptable quality in multicultural populations, and is a means for efficient modification of the educational process of nurses based on this assessment.

### Purpose

The goal of this article is to offer a preliminary assessment of the cultural competence of nurses based on a Polish-language and -culture version of the Nurse Cultural Competence Scale (NCCS).

### Research method

An adaptive and diagnostic cross-disciplinary concept was used in the research. Two hundred thirty-eight professionally active nurses in the southeast region of Poland took part in this study. The NCCS-Polish version (NCCS-P) questionnaire was used after linguistic adaptation and analysis of psychometric properties.

### Results

Moderate levels of competence in the Cultural Knowledge Subscale (M = 3.42) were found in the group of nurses studied. The results indicate lowest competency levels in the Cultural Skill Subscale (M = 3.14). The highest values were obtained for the Cultural Awareness Subscale (M = 3.98) and the Cultural Sensitivity Subscale (M = 3.72). The Cronbach's alpha reliability coefficient for the NCCS-P scale was 0.94, with the subscale values ranging from 0.72 to 0.95. Factor validity analysis of the Polish adaptation of the NCCS-P scale pointed to its four-factor structure. The Kaiser-Mayer-Olkin sampling adequacy test was 0.905, and

**Funding:** The publication was financed with funding from the Medical University No. DS. 514 and No. DS.519. The funders had no role in study design, data collection and analysis, decision to publish, or preparation of the manuscript.

**Competing interests:** The authors have declared that no competing interests exist.

the Bartlett test of sphericity result was $\chi2 = 5755.107$; df = 820; p<0.001. The four-factor structure is affirmed by the Kaiser criterion and the scree test result.

## Conclusions

The NCCS-P psychometric properties were highly reliable and significant because of the opportunity for using them for research in Poland.

## Practical implications

The scale can be used in intercultural research for comparing cultural competence of nurses, including Polish ones. This scale facilitates the precise monitoring of cultural competence among nurses and nurse managers, which may help in developing nursing policies geared toward a commitment to expanding cultural competence.

## Introduction

Over a long period of its development (the post-war period of the 20[th] century), Polish society was seen as culturally monolithic. The occurrence of multiculturality became noticeable only after 1989. Attention was then drawn to the existence of historical ethnic and national minorities as well as to the influx of new communities (refugees, immigrants or repatriates) [1].

The process of globalization as well as the development of the free market are also developing cultural diversity in monocultural Poland, and with it the dominance in public life of the principle of competition, which leads to diversity not only of consumer goods. Global migration trends indicate that migration is on the increase and that a steady growth in the number of people migrating to Poland can be expected. In 2014 the number of persons coming into Poland (for temporary or permanent residence) was about 64,000, while in 2018 that number increased by four-fold to over 268,000 [2]. The vast majority of the incoming population were citizens of Ukraine, followed by Byelorussia, Germany, and Vietnam. Migrants to Poland tend to be aged 20–39 years old, and are most often men. The majority of these people are foreign students who have come to study in Polish universities [2].

Together with this influx of foreigners, new challenges arrive in providing healthcare: in communication, acceptance of cultural differences, and a disproportion in healthcare knowledge among migrants. There are a number of problems related to providing adapted and culturally diverse care by healthcare professionals [3, 4].

The Polish National Census conducted in 2011 showed more than 97.1% of respondents declaring Polish nationality, 1.55% belonging to other nationalities, and 1.35% of the population not specifying a nationality. The majority of the Polish population identifies with religious institutions (88.9%) [5].

The religious homogeneity of Polish society is an additional factor influencing the specificity of the social situation of immigrants. In 2011 there were over 34 million Christians in Poland, forming 88.8% of the population [6], although the religiosity of young adult Poles is on the decline [7]. This situation leads to the necessity of having, and therefore of forming, cultural competencies of healthcare workers [8] such as those in other countries [9, 10]. Poland ratified the UNESCO Convention on the Protection and Promotion of the Diversity of Cultural Expressions in 2007, which makes possible an intensification of efforts to develop cultural competencies [11].

## Forming cultural competence of nurses

The formation of professional competence takes place at every level of undergraduate and postgraduate training, and is also regulated by legal and ethical norms. In accordance with the act of July 15, 2011 on the nursing and midwifery professions, Polish nurses qualify as general nurses after completing [nursing] college. A significant piece of legislation, in which explicit guidelines for formation in cultural competencies for nursing students were included for the first time in 100+ years of nursing training [12], is the Regulation of the Minister of Science and Higher Education concerning training standards for the fields of study of: medicine, dental medicine, pharmacology, nursing, and midwifery, which regulates training outcomes to be achieved by students [13].

In an analysis of this document's contents in the area of transcultural care, educational effects can be distinguished which emphasize cognizance of cultural and religious separateness, as well as interpretation of phenomena of class, ethnic and gender inequality and discrimination; knowledge on the topic of the health insurance system in Poland and the European Union; knowledge of the content of the Charter of Human Rights; recognition of cultural, social and economic determinants of public health; and analysis and critical assessment of discrimination and racism. An analysis of the cultural competence possessed by nursing students and Polish nurses [8] indicates a lack of knowledge on the topic of a given culture as a significant barrier, hindering relationships with patients from other cultural backgrounds and the emergence of positive reactions, chiefly openness and understanding when assessing a person coming from a different culture.

Therefore, as of 2019, content on cultural competence has been expanded in the training standards for the field of nursing. The additions consist of an emphasis on training results in the area of cultural competence to be achieved by all graduates of the first cycle of nursing studies, i.e.: knowledge of cultural public health determinants, and will also propose measures preventing discrimination and racism among children and youth.

Multicultural Nursing, appearing for the first time as part of the legal regulations for second-cycle nursing studies, is now a required course. Its anticipated effects on graduates' knowledge and skills include: cultural conditioning in providing care, including health behaviors and treatment approaches; awareness of cultural and religious differences in perceptions and in intercultural communication; making use of differences in interpersonal communication arising from cultural, ethnic, religious and social conditions; applying in practice the principles of the Madeleine Leininger theory of multicultural nursing; recognizing cultural nutritional and transfusiological conditions; and taking into account patients' religious and cultural conditions in their healthcare needs [14].

Another important document in nursing practice is the Code of Professional Ethics for Nurses and Midwives of the Republic of Poland. This Code refers to transcultural care in its preamble, that nurses show respect to patients and give aid to each person regardless of race, creed, nationality, political viewpoint, economic status, or other differences. This declaration also forms the content of the Nursing Pledge made when receiving the diploma for a bachelor's degree in nursing. This document also identifies the role of nurse as a person who, at the patient's request, takes care of contacting a clergyman and ensures dignified conditions for death, along with respecting the values held by the patient [15].

For social competence in postgraduate education, it is advised that nurses specializing in various fields respect the patient's dignity and autonomy regardless of his or her age, gender, disability, sexual orientation, or national and ethnic origin, where as the humanities curriculum emphasizes the importance of knowledge in the area of the values systems, religious beliefs, and customs of patients of different nationalities [16].

During these changes, the choice of methods for forming cultural competences [17], which will be based on the Campinha-Bacote concept using a value-based approach [18], should be considered important.

The profile for formal conditions determining the scope of formation in, and implementation of, cultural competences tends toward adopting clear theoretical objectives and adequate tools for measuring cultural competence in order to create a methodologically appropriate environment for monitoring their development, which largely determines the subsequent effects and consequences of their implementation [19].

Currently, for the purposes of testing cultural competence of the Polish nursing community, several scales have been adapted, i.e. the Cultural Intelligence Scale (CQS) by Ang, Van Dyne and Koh, which makes it possible to estimate cultural intelligence, that is to say, the individual's ability to deal effectively in situations of cultural diversity [20, 21]. This same team has adapted and assessed the psychometric values of The Cross-Cultural Competence Inventory, permitting an evaluation of the seven dimensions of intercultural competence, i.e.: cultural adaptability, self-presentation, tolerance of uncertainty, determination, engagement, mission focus, lie and social desirability scale [22]. A third validated scale for assessing cultural competence of Polish nurses is the Healthcare Provider Cultural Competence Instrument (HPCCI) by the team of Schwarz et al. [23]. This scale was validated in the international project Multicultural Care in European Intensive Care Units [24].

## Background

Lack of cultural knowledge and skills in nurses can contribute to the development of difficulties in building therapeutic relationships with patients and lead to inequality in the provision of care [25]. Transcultural care allows for diverse means of holistic, individualized, and humane healthcare [26]. Over the years, the need to shape cultural competence has brought about the creation of transcultural care models [27]. Such models include, among others: the Rising Sun Model by M. Leininger [28, 29]; the L. Purnell Cultural Competence Model [30], The Process of Cultural Competence in the Delivery of Healthcare Services Model of J. Campinha-Bacote [31] and the Transcultural Assessment Model by J. Giger and R. Davidhizar [32]. These models have given rise to the development of many tools for assessing levels of cultural competence.

Each author's concept of cultural competence takes a slightly different approach; some authors focus on the component of competence as a process or final goal, and some on a cultural factor which might be based on values, on convictions about health, or on religion or philosophy, taking into account the cultural group as well as the patient as an individual. Each author concentrates on various attributes of cultural competence. For example, Campinha-Bacote concentrates on such characteristics of cultural competence as knowledge, attitude, and abilities [33], while Leininger or Giger and Davidhizar address themselves to such aspects as cultural values, religion, and convictions about health [29, 32]. For Purnell and Paulanka [34] cultural competence is a set of elements: cultural self-awareness, knowledge and understanding of the culture of the client, respect for cultural differences, openness to cultural encounter, and adjustment of care based on culture. The characteristics most often appearing in the various concepts of cultural competence are: cultural awareness, cultural knowledge, cultural sensitivity, cultural skills, cultural proficiency, and dynamics [35].

Cultural awareness is the process of making an independent assessment of the impact of one's own culture on other cultures and of recognizing the similarities and differences between other cultures and one's own culture. Cultural awareness is the foundation for assessing the beliefs and values of others [33, 36–38].

Cultural knowledge is the process by which the healthcare professional seeks and obtains a thorough education on cultural diversity, which helps him in analyzing data. In this process, the healthcare professional should integrate the aspect of [healthcare] practice with health-related and cultural values as well as with the epidemiology of diseases and effectiveness of their treatment [33, 35, 39].

Cultural skills are manifested in the ability to carry out cultural evaluations in order to gather appropriate cultural information needed for ensuring care; these are skills for establishing effective communication with persons of other cultures [40, 41].

Cultural encounter is the process by which a nurse takes direct part in cultural interaction with patients from culturally diverse environments in order to adjust his or her own beliefs about a cultural group and to prevent possible stereotyping [31, 33, 38].

Cultural sensitivity is the ability to show respect and appreciation for cultural diversity. It is the attempt to understand the world the way a person from another culture perceives it. This characteristic helps nurses to understand how the attitudes and points of view of patients affect their behavior and standards in seeking healthcare [42–45].

Cultural proficiency is the ability to acquire and communicate new information by conducting research and sharing results through articles, educational programs, and other media [35, 46].

Cultural dynamicity is the acquiring of cultural competence through frequent meetings with various patients of different cultures [47, 48].

Taking the various theories of cultural competence into consideration, we were looking for a tool based on the Campinha-Bacote concept due to its extensiveness and the cultural competencies encompassed in this model, as well as their being defined as a process by which the nurse continually strives to gain effective patient-care skills in the area of culture [49]. In the first version of the Campinha-Bacote model of 1991, four components of cultural competence were identified. These were cultural awareness, cultural knowledge, cultural skills, and cultural encounter. In 1998 the author of this model added a fifth construct called cultural desire as a completion to the model. The current model begins and ends with seeking and experiencing cultural encounter and only by continual cultural encounter will cultural awareness, cultural knowledge, cultural skills, and cultural desire be obtained [31, 38, 50].

The Nurse Cultural Competence Scale (NCCS) by Perng and Watson was developed and delineated using Mokken scaling. This model was created with reference to the Campinha-Bacote model for healthcare cultural competence. Several studies have confirmed the reliability and accuracy of this tool. The scale consists of 41 items for measuring four constructs: cultural awareness (items 1–10); cultural knowledge (items 11–19); cultural sensitivity (items 20–27); and cultural skills (items 28–41). The Cronbach's alpha result for the four constructs ranged from 0.78 to 0.96 [51].

Cultural competences are very specific and individual abilities, without which nurses cannot develop holistic, culturally-adapted care plans in cases of culturally diverse patients, but only plans based on a uniform standard. That is why in professional practice a simple to use, accurate, and reliable scale of cultural competences is needed to assess nurses and their interactions with culturally and/or ethnically diverse patients.

## Purpose of the study

The purpose of this work is to present preliminary results of a cultural competence assessment of nurses in Poland, based on an adaptation of the NCCS scale, and its connection to the seniority and work experience of nurses in caring for patients of different cultures.

## Methods

### Setting

The study used an adaptive-diagnostic cross-sectional project. A sampling of 238 professionally actives nurses was recruited in the southeastern region of Poland. These nurses represented nursing experience in hospital departments as well as in out-patient healthcare.

### Procedure

The procedure for adapting the scale was based on the process of translation and adaptation of instruments recommended by the WHO [52]. As part of the adaptation procedure of the original version, the NCCS was translated into Polish by two independent translators. Next, a panel of experts made up of nurses of varying professional experience and working in various positions, developed version 1.0 of the NCCS scale, which was re-translated into English by an independent translator who was a native speaker of English. After a comparison of the original and adapted 1.0 versions, a pilot study was conducted with 15 nurses. The interviewer, who was one of the authors of the document, held discussions with the pilot study participants on the clarity and comprehensibility of the survey items, which made it possible to form the final version of the NCCS, which was subjected to the further examinations presented in this article. Every stage of the language adaptation of the scale has been documented. The study used the Nurse Cultural Competence Scale (NCCS) questionnaire of S. Perng and R. Watson [51], who have given written validation and consent for use of the scale in Poland. Based on the J. Campinha-Bacote model of cultural competence in providing healthcare, we made a cultural adaptation of the Nurse Cultural Competence Scale with the aid of the translation-back translation procedure [53]. Guidelines for translating, adapting and validating the scale [53] were also observed to maintain semantic, idiomatic and conceptual equivalence to the original questionnaire.

Two bilingual experts and two Polish nurses translated the questionnaire content from English to Polish. The translations were then compared and the main inconsistencies identified and discussed. A unique Polish version of the NCCS questionnaire was developed. The Polish questionnaire was then translated into English by another bilingual expert in order to assess its equivalence. The translated version was next compared with the original questionnaire. A comparison of the wording of items on the scale of their significance found them to be clear and eminently equivalent. A final version of the Polish instrument having been obtained, preliminary test was conducted with the participation of 20 nurses to ensure the continued validity of the adapted questionnaire. The initial test assessed the quality of the translation, appropriate cultural adaptation, and the possibility of putting the instrument into practice. Moreover, it allowed the researchers to assess the amount of time needed for filling out the questionnaire (i.e. 20 minutes). The nurses were invited to submit written recommendations for improving comprehensibility of the scale items as well as for improving the graphic structure. Some minimal alterations of content and graphics were made after this preliminary test.

### Participants and data collection

The study was conducted on the premises of hospitals, clinics, and nursing homes in the Lublin Voivodeship, which lies in southeastern Poland (Central Europe), inhabited by a Slavic population of Caucasian race and Roman Catholic religion. Most of those registered for a longer than three-month residence in the Lublin Voivodeship are from Ukraine: 3310 persons (63.0%), 2812 (85%) of whom are short-term immigrants [54].

All nurses interested in taking part in the study were provisionally recruited. Criteria for inclusion were: being an active professional nurse; currently working full- or part-time in public healthcare. Nurses working in managerial positions were excluded from the study since they did not have direct interaction with patients. Additionally, student nurses were excluded because they are not formally recognized as Registered Nurses, and their cultural competence is still being formed as part of their university education.

The research toolkit was prepared in printed form and a total of 300 questionnaires were distributed to nurses during working hours, placed in addressed envelopes with information on how to return the questionnaire. The data were collected from May to October, 2018.

## Instruments

The Polish version of the Nurse Cultural Competence Scale (NCCS-P) was used to evaluate cultural competences among professionally active nurses. The original version of the NCCS has 41 items, forming four subscales: Cultural Awareness (NCCS-CA-P)-10 items, Cultural Knowledge (NCCS-CK-P)-9 items, Cultural Sensitivity (NCCS-CSe-P)-8 items, and Cultural Skills (NCCS-CS-P)-14 items. Respondents' answers are assigned points on a 5-point Likert scale, determining the degree of agreement with indicated statements, going from strongly disagree (1) to strongly agree (5). The overall score falls within a range of 41 to 205 points. The higher the score, the higher is the level of cultural competence.

## Ethical considerations

The study was authorized by the Bioethical Commission of the Medical University of Lublin (No. KE-0254/307/2018) and was also in accordance with the ethical principles of the Declaration of Helsinki. Each participant received oral and written information about the purpose of the study. Participation was voluntary and anonymous. All the nurses were informed that they could withdraw from completing the survey at any time, without consequences and without having to give reasons. For the sake of preserving anonymity, the research tools were not given identifying marks. Returning the completed questionnaire was considered as equivalent to giving informed consent to participation in the study.

## Data analysis

The statistical characteristics of continuous variables are presented as arithmetic means and their standard deviations (SD). The distribution of continuous variables was verified by the Shapiro-Wilk test. The distributions of discrete variables were presented in the form of numbers and percentages. The Pearson R and the Spearman Rho correlation coefficients were used to assess the relationship between variables. The Kruskal-Wallis nonparametric test was used to determine intergroup differences for more than two independent groups.

Cronbach's alpha coefficient was used to assess reliability using the internal consistency test method. The Kaiser-Mayer-Olkin test was used to test the suitability of the sample selection. Theoretical validity was assessed using exploratory factor analysis, which was carried out by method of the main components of the simple Oblimin rotation and Kaiser normalization. The reliability of the tool was assessed based on the discriminant power values of items forming the featured dimensions.

A significance level of $p<0.05$ was adopted, indicating the occurrence of statistically significant differences or dependencies. Statistical analysis was performed using IBM SPSS Statistics (PS IMAGO) v. 25 software.

## Results

### Characteristics of the participants

A total of 300 questionnaires were distributed. The response rate recorded at the end of the analysis, based on missing fully-completed questionnaires, was 79.33%. Therefore, the overall sample included 238 nurses. The study group was predominantly made up of women, who formed 93.7% of respondents.

The age range of those studied was from 20 to 56 years old (M = 32.68; SD = 10.74 years), and length of service from six months to 36 years (M = 9.32; SD = 10.19 years). Over 51.7% of the nurses had experience in caring for patients from other cultures. The specific characteristics of the study group are given in Table 1.

### Assessment of the relevance of the factors of the Polish adaptation NCCS-P scale

The factor structure for analyzing a scale, as well as many other tools, depends on conceptual conditions, cultural meanings of the ideas being analyzed, type of sample, statistical procedures used, factor criteria, etc.

**Table 1. Characteristics of the group of nurses studied.**

| Socio-demographic and professional variables | | N | % |
|---|---|---|---|
| Sex | Female | 223 | 93.7 |
| | Male | 15 | 6.3 |
| Education | Registered Nurse | 2 | 0.8 |
| | Bachelor of Nursing | 234 | 98.3 |
| | Masters in Nursing | 2 | 0.8 |
| Age | 20–29 | 118 | 49.6 |
| | 30–39 | 39 | 16.4 |
| | 40–49 | 64 | 26.9 |
| | 50–56 | 17 | 7.1 |
| Job seniority | Up to one year | 63 | 26.5 |
| | 1–5 years | 64 | 26.9 |
| | 6–10 years | 20 | 8.4 |
| | 11–20 years | 57 | 23.9 |
| | 21–30 years | 27 | 11.3 |
| | Over 30 years | 7 | 2.9 |
| Place of residence | Village | 127 | 53.4 |
| | Small town | 71 | 29.8 |
| | Large city | 40 | 16.8 |
| Care for patients of different culture | Yes | 123 | 51.7 |
| | No | 89 | 37.4 |
| | Don't know | 26 | 10.9 |
| Marital status | Single | 100 | 42.0 |
| | In a relationship | 138 | 58.0 |
| Place of work | Hospital ward | 120 | 50.4 |
| | Primary Health Care | 41 | 17.2 |
| | Home Social Welfare | 23 | 9.7 |
| | Care and Treatment Facilities | 28 | 11.8 |
| | Outpatient specialist care | 26 | 10.9 |

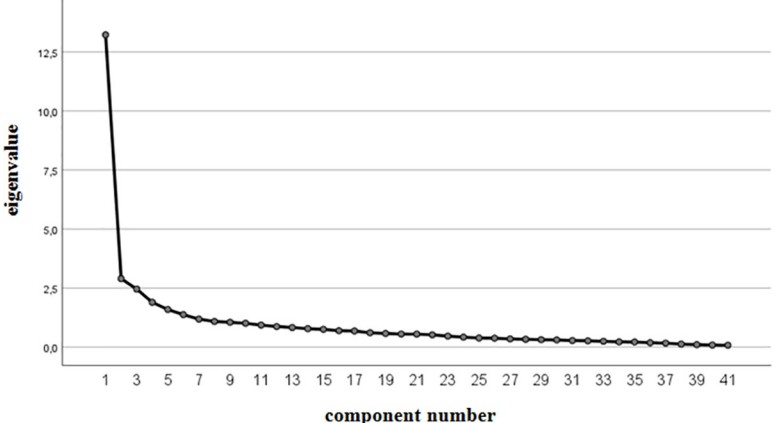

**Fig 1. Scree plot of the psychometric properties of the Polish adaptation of the NCCS scale.**

This study undertakes examining the factor relevance of the Polish adaptation of the NCCS-P scale by use of factor analysis (main component model, Oblimin rotation).

With an unforced number of factors, 10 eigenvalues greater than 1.0 were obtained. They account for a total of 67.73% volatility. This is not a satisfactory result for confirming a theoretical model. Therefore, a 4-factor solution was forced, in order to see whether it would be possible to reconstruct the subscales and the affiliation of the individual items of the Polish NCCS adaptation to the original subscales.

The 4-factor solution proved to be satisfactory. The value for the Kaiser-Mayer-Olkin sample adequacy test is 0.905, and a strong result was obtained for the Bartlett sphericity test ($\chi 2$ = 5755.107; df = 820; p<0.001). Both the Kaiser criterion (four loadings above the value of 1) and the result of the scree plot test justify the adoption of this scale structure (Fig 1).

Analysis revealed in most cases medium and high values for factor loadings of all items constituting individual subscales, with the exception of item n. 28 (with a borderline score of 0.298). It was decided, however, not to delete this item. The values for factor loadings of the items making up the featured dimensions were: for the first factor NCCS-CS-P from 0.298 to 0.924; for the second factor NCCS-CA-P from 0.304 to 0.740; for the third NCCS-CSe-P from 0.379 to 0.703; and for the fourth NCCS-CK-P from 0.387 to 0.729 (Table 2).

The first factor with an intrinsic value of 13.23 accounts for 32.26% of the total variance and is correlated with 14 items (statements 28–41), according to the original version of the NCCS. The second factor, with an eigenvalue of 2.90, accounting for 7.07% of variance, consists of 9 items (statements 1 through 9 for this subscale do not coincide with the original version), and one item less than the original version of the tool. The third factor, with an eigenvalue of 2.46, accounting for 6.00%, brings together 8 items (statements 20–28) (a reconstruction of the original subscale design). The fourth factor with an eigenvalue of 1.90, accounts for 4.62% of variance and is composed of 10 items (9 of the items in the Polish version are covered in the original version; statement 10 has been added). The Polish version retains the original names for the individual subscales (Table 3).

## Assessment of the reliability of the Polish adaptation of the NCCS-P scale

The value of the Cronbach $\alpha$ coefficient for the whole scale, composed of 41 items, is 0.94. The Cronbach $\alpha$ value for the first factor NCCS-CS-P is 0.95; the second factor NCCS-CA-P-0.72; for the third NCCS-CSe-P-0.77, and for the fourth NCCS-CK-P-0.87. All of the results fulfill

**Table 2. Values for factor loadings of items included in the separated scales of the NCCS-P questionnaire.**

| Subscales / items | NCCS-CS-P | NCCS-CA-P | NCCS-CSe-P | NCCS-CK-P |
|---|---|---|---|---|
| Item 28 | 0.298 | -0,101 | 0,272 | 0,223 |
| Item 29 | 0.311 | -0,014 | 0,224 | 0,130 |
| Item 30 | 0.364 | 0,139 | 0,284 | 0,054 |
| Item 31 | 0.419 | 0,008 | 0,279 | 0,153 |
| Item 32 | 0.429 | -0,063 | 0,263 | 0,220 |
| Item 33 | 0.434 | -0,050 | 0,224 | 0,233 |
| Item 34 | 0.624 | -0,033 | 0,209 | 0,170 |
| Item 35 | 0.585 | 0,033 | 0,177 | 0,117 |
| Item 36 | 0.338 | -0,069 | 0,234 | -0,019 |
| Item 37 | 0.931 | 0,061 | -0,044 | -0,055 |
| Item 38 | 0.942 | 0,053 | -0,152 | 0,025 |
| Item 39 | 0.943 | 0,052 | -0,052 | -0,017 |
| Item 40 | 0.896 | -0,048 | -0,059 | 0,104 |
| Item 41 | 0.881 | -0,004 | -0,105 | 0,087 |
| Item 1 | 0,146 | 0.304 | 0,026 | -0,024 |
| Item 2 | 0,197 | 0.397 | -0,243 | 0,045 |
| Item 3 | 0,065 | 0.321 | -0,220 | 0,226 |
| Item 4 | 0,047 | 0.670 | 0,016 | 0,011 |
| Item 5 | -0,195 | 0.633 | -0,013 | 0,049 |
| Item 6 | 0,012 | 0.705 | 0,094 | -0,022 |
| Item 7 | 0,024 | 0.551 | 0,138 | -0,116 |
| Item 8 | 0,021 | 0.740 | 0,064 | -0,070 |
| Item 9 | 0,009 | 0.498 | 0,007 | 0,116 |
| Item 20 | 0,258 | 0,216 | 0.480 | 0,094 |
| Item 21 | -0,018 | 0,078 | 0.447 | 0,004 |
| Item 22 | -0,018 | 0,084 | 0.703 | 0,114 |
| Item 23 | 0,045 | 0,015 | 0.379 | 0,056 |
| Item 24 | -0,052 | 0,151 | 0.409 | -0,060 |
| Item 25 | 0,186 | 0,146 | 0.457 | 0,144 |
| Item 26 | 0,086 | 0,176 | 0.534 | 0,282 |
| Item 27 | -0,016 | 0,095 | 0.548 | 0,122 |
| Item 10 | -0,119 | 0,124 | -0,230 | 0.564 |
| Item 11 | -0,060 | 0,309 | 0,055 | 0.387 |
| Item 12 | -0,001 | -0,078 | 0,137 | 0.545 |
| Item 13 | 0,124 | -0,023 | 0,003 | 0.714 |
| Item 14 | 0,122 | 0,066 | 0,063 | 0.557 |
| Item 15 | -0,010 | -0,011 | 0,040 | 0.706 |
| Item 16 | 0,073 | -0,061 | 0,117 | 0.729 |
| Item 17 | 0,149 | -0,107 | 0,145 | 0.651 |
| Item 18 | 0,239 | -0,034 | -0,015 | 0.639 |
| Item 19 | 0,196 | -0,163 | 0,185 | 0.617 |

* Principal component method with Oblimin simple rotation and Kaiser normalization.

NCCS-CA-P: Cultural Awareness Subscale; NCCS-CK-P: Cultural Knowledge Subscale; NCCS-CSe-P: Cultural Sensitivity Subscale; NCCS-CS-P: Cultural Skills Subscale

**Table 3. Polish adaptation of the NCCS scale-discriminant power indicators and Cronbach's α coefficient.**

| No. | Subscales | Original version | Polish adaptation | Discriminatory powers | Cronbach's alpha |
|---|---|---|---|---|---|
| 1. | NCCS-CS-P | 14 items | 14 items | 0.496–0.819 | 0.95 |
| 2. | NCCS-CA-P | 10 items | 9 items | 0.211–0.540 | 0.72 |
| 3. | NCCS-CSe-P | 8 items | 8 items | 0.273–0.645 | 0.77 |
| 4. | NCCS-CK-P | 9 items | 10 items | 0.313–0.717 | 0.87 |
| | | | | Total NCCS-P | 0.94 |

NCCS-CA-P: Cultural Awareness Subscale; NCCS-CK-P: Cultural Knowledge Subscale; NCCS-CSe-P: Cultural Sensitivity Subscale; NCCS-CS-P: Cultural Skills Subscale

the Nunnally criterion (α Cronbach coefficient >0.07). The discriminant power index for NCCS-CS-P was within a range of 0.496–0.819; for NCCS-CA-P: 0.211–0.540; for NCCS-CSe-P: 0.273–0.645; and for NCCS-CK-P: 0.313–0.717 (Table 3).

## Characterization of cultural competence in the nurses studied

The cultural competence of the nurses studied using the Polish version of the NCCS-P, consisting of 41 items forming four subscales, achieved an overall result of a range of 93 to 192 points. The range of values of cultural competences has a normal distribution. Cultural competence of nurses' cultural awareness (NCCS-CA-P), assessed with 9 items, took a range of results from 25 to 45 (M = 35.84). The cultural knowledge (NCCS-CK-P) of the nurses studied, estimated on a basis of 10 items, reached a range of results from 18 to 48 (M = 34.28). Meanwhile, cultural sensitivity (NCCS-CSe-P), assessed by 8 items, obtained a range of results from 17 to 40 (M = 29.83). Cultural Skills (NCCS-CS-P), assessed on the basis of 14 items, took a value of 20 to 70 (M = 43.97). The average unit overall score of cultural competences (NCCS-P) in the study group was 143.92 (SD = 20.07). The constituent values of the nurses' cultural competences were normal (Table 4).

Moderate levels of competence in the NCCS-CK Subscale (M = 3.42) were found in the nurses studied. The results indicate the lowest level of competence in the NCCS-CS Subscale (M = 3.14). The highest values obtained by the study group were in the NCCS-CA Subscale (M = 3.98) and the NCCS-CSe Subscale (M = 3.72) (Table 5).

The cultural competences of the nurses studied, analyzed based on a detailed record of items, were quite diverse. They delineate certain trends in cultural skills. The nurses did not consider themselves as competent for being sources of information for other nurses in terms of cultural competence in practice (item 37: M = 2.80, SD = 1.00; item 38: M = 2.82, SD = 1.00;

**Table 4. Descriptive statistics and Shapiro-Wilk test values for the NCCS-P scale and its subscales.**

| Subscales | | | M | SD | Min | Max | S-W | |
|---|---|---|---|---|---|---|---|---|
| | Original version | Polish adaptation | | | | | Results | p |
| NCCS-CA-P | 10 items | 9 items | 35.84 | 4.21 | 25.00 | 45.00 | 0.987 | 0.026 |
| NCCS-CK-P | 9 items | 10 items | 34.28 | 5.75 | 18.00 | 48.00 | 0.991 | 0.165 |
| NCCS-CSe-P | 8 items | 8 items | 29.83 | 4.63 | 17.00 | 40.00 | 0.985 | 0.011 |
| NCCS-CS-P | 14 items | 14 items | 43.97 | 10.70 | 20.00 | 70.00 | 0.991 | 0.153 |
| Total score:NCCS-P | | | 143.92 | 20.07 | 93.00 | 192.00 | 0.990 | 0.105 |

NCCS-CA-P: Cultural Awareness Subscale; NCCS-CK-P: Cultural Knowledge Subscale; NCCS-CSe-P: Cultural Sensitivity Subscale; NCCS-CS-P: Cultural Skills Subscale; M-mean; SD-standard deviation; S-W- Shapiro-Wilk test of normality

**Table 5. Detailed NCCS-P characteristics of cultural competence of the nurses surveyed.**

| | | | | | | | |
|---|---|---|---|---|---|---|---|
| **NCCS-CS-P** | | | | | | | |
| **Item number** | M | SD | Min | Max | Percentile | | |
| | | | | | 25 | 50 | 75 |
| **28** | 3.27 | 0.99 | 1.00 | 5.00 | 3.00 | 3.00 | 4.00 |
| **29** | 3.29 | 1.04 | 1.00 | 5.00 | 3.00 | 3.00 | 4.00 |
| **30** | 3.21 | 1.21 | 1.00 | 5.00 | 2.00 | 3.00 | 4.00 |
| **31** | 3.26 | 1.00 | 1.00 | 5.00 | 3.00 | 3.00 | 4.00 |
| **32** | 3.17 | 0.93 | 1.00 | 5.00 | 3.00 | 3.00 | 4.00 |
| **33** | 3.23 | 0.95 | 1.00 | 5.00 | 3.00 | 3.00 | 4.00 |
| **34** | 3.18 | 0.96 | 1.00 | 5.00 | 3.00 | 3.00 | 4.00 |
| **35** | <u>3.42</u> | 0.96 | 1.00 | 5.00 | 3.00 | 4.00 | 4.00 |
| **36** | <u>3.65</u> | 0.89 | 1.00 | 5.00 | 3.00 | 4.00 | 4.00 |
| **37** | <u>2.80</u> | 1.00 | 1.00 | 5.00 | 2.00 | 3.00 | 3.00 |
| **38** | 2.82 | 1.00 | 1.00 | 5.00 | 2.00 | 3.00 | 3.00 |
| **39** | 2.83 | 0.99 | 1.00 | 5.00 | 2.00 | 3.00 | 4.00 |
| **40** | 2.86 | 1.00 | 1.00 | 5.00 | 2.00 | 3.00 | 4.00 |
| **41** | 2.97 | 1.04 | 1.00 | 5.00 | 2.00 | 3.00 | 4.00 |
| **Subscale NCCS-CS-P** | **3.14** | | - | - | - | - | - |
| **NCCS-CA-P** | | | | | | | |
| **Item number** | M | SD | Min. | Max | Percentile | | |
| | | | | | 25 | 50 | 75 |
| **1** | <u>4.12</u> | 0.74 | 2.00 | 5.00 | 4.00 | 4.00 | 5.00 |
| **2** | <u>3.65</u> | 0.92 | 1.00 | 5.00 | 3.00 | 4.00 | 4.00 |
| **3** | 3.87 | 0.83 | 1.00 | 5.00 | 3.00 | 4.00 | 4.00 |
| **4** | 4.19 | 0.86 | 1.00 | 5.00 | 4.00 | 4.00 | 5.00 |
| **5** | 3.94 | 0.78 | 2.00 | 5.00 | 3.00 | 4.00 | 4.50 |
| **6** | 4.01 | 0.83 | 1.00 | 5.00 | 4.00 | 4.00 | 5.00 |
| **7** | 3.72 | 0.92 | 1.00 | 5.00 | 3.00 | 4.00 | 4.00 |
| **8** | 4.07 | 0.88 | 1.00 | 5.00 | 3.00 | 4.00 | 5.00 |
| **9** | <u>4.30</u> | 0.77 | 2.00 | 5.00 | 4.00 | 4.00 | 5.00 |
| **Subscale NCCS-CA-P** | **3.98** | | - | - | - | - | - |
| **NCCS-CSe-P** | | | | | | | |
| **Item number** | M | SD | Min | Max | Percentile | | |
| | | | | | 25 | 50 | 75 |
| **20** | 3.73 | 0.83 | 1.00 | 5.00 | 3.00 | 4.00 | 4.00 |
| **21** | 3.84 | 0.99 | 1.00 | 5.00 | 3.00 | 4.00 | 5.00 |
| **22** | <u>3.91</u> | 0.85 | 2.00 | 5.00 | 3.00 | 4.00 | 5.00 |
| **23** | 3.47 | 0.94 | 1.00 | 5.00 | 3.00 | 3.00 | 4.00 |
| **24** | <u>3.48</u> | 0.91 | 1.00 | 5.00 | 3.00 | 4.00 | 4.00 |
| **25** | 3.81 | 0.96 | 1.00 | 5.00 | 3.00 | 4.00 | 4.00 |
| **26** | 3.81 | 0.93 | 1.00 | 5.00 | 3.00 | 4.00 | 4.00 |
| **27** | 3.80 | 1.10 | 1.00 | 5.00 | 3.00 | 4.00 | 5.00 |
| **Subscale NCCS-CSe-P** | **3.72** | | - | - | - | - | - |
| **NCCS-CK-P** | | | | | | | |
| **Item number** | M | SD | Min | Max | Percentile | | |
| | | | | | 25 | 50 | 75 |
| **10** | 3.52 | 0.92 | 1.00 | 5.00 | 3.00 | 3.00 | 4.00 |
| **11** | <u>3.93</u> | 0.70 | 2.00 | 5.00 | 4.00 | 4.00 | 4.00 |

*(Continued)*

**Table 5.** (Continued)

| | | | | | | | |
|---|---|---|---|---|---|---|---|
| 12 | 3.57 | 0.75 | 2.00 | 5.00 | 3.00 | 4.00 | 4.00 |
| 13 | 3.23 | 0.84 | 1.00 | 5.00 | 3.00 | 3.00 | 4.00 |
| 14 | 3.37 | 0.86 | 1.00 | 5.00 | 3.00 | 3.00 | 4.00 |
| 15 | <u>3.65</u> | 0.89 | 1.00 | 5.00 | 3.00 | 4.00 | 4.00 |
| 16 | 3.28 | 0.87 | 1.00 | 5.00 | 3.00 | 3.00 | 4.00 |
| 17 | 3.32 | 0.85 | 1.00 | 5.00 | 3.00 | 3.00 | 4.00 |
| 18 | <u>3.16</u> | 0.90 | 1.00 | 5.00 | 3.00 | 3.00 | 4.00 |
| 19 | 3.25 | 0.93 | 1.00 | 5.00 | 3.00 | 3.00 | 4.00 |
| **Subscale NCCS-CK-P** | **3.42** | | - | – | - | - | - |

NCCS-CA-P: Cultural Awareness Subscale; NCCS-CK-P: Cultural Knowledge Subscale; NCCS-CSe-P: Cultural Sensitivity Subscale; NCCS-CS-P: Cultural Skills Subscale.

Items based on: Perng SJ, Watson R. Construct validation of the Nurse Cultural Competence Scale: a hierarchy of abilities. J Clin Nurs. 2012;21(11–12): 1678–1684. doi: 10.1111/j.1365-2702.2011.03933.x.

item 39: M = 2.83, SD = 0.99), but were at the same time sure that they were able to meet the needs of patients in accordance with the patients' preferences as conditioned by cultural factors (item 36: M = 3.65, SD = 0.89; item 35: M = 3.42, SD = 0.96).

Those being researched attained the highest values of cultural competence in the area of cultural awareness. They expressed the opinion that nursing education has an enormous influence in forming cultural competence and in understanding patients' cultural backgrounds (item 9: M = 4.30, SD = 0.77). In the area of cultural sensitivity, the highest value determined was for the ability to tolerate beliefs or behavior of various cultural groups regarding health/illness (item 22: M = 3.91, SD = 0.85). On the other hand, the lowest value in this group of items was for the statement about the acceptance of the client's health maintenance method, even if it is different from the professional knowledge of the nurse (item 24: M = 3.48, SD = 0.91). In the area of cultural knowledge, the items concerning knowledge of facts were rated quite high (item 11: M = 3.93, SD = 0.70; item 15: M = 3.65, SD = 0.89), yet in comparison, cultural knowledge is reported with lower values (item 18: M = 3.16, SD = 0.90) (Table 5).

In addition, correlations of the NCCS-P scale with age, seniority, and experience of culturally different patient care were also determined. It was determined that cultural awareness increases along with both the age and the professional experience (seniority) of those studied. A higher level of cultural competence was noted from those respondents in the study group who had experienced contact with a patient from another culture, in relation both to the overall result and to all of the analyzed subscales (Table 6).

## Discussion

### Psychometric validation of the scale

The NCCS scale is an instrument designed to measure the ability of nurses to provide holistic care for culturally different recipients. The Polish version, the NCCS-P, like the original version, has a 4-factor structure which meets the criteria for theoretic validity. The NCCS scale has so far been validated in a few countries where the process of developing the cultural competence of nurses is recognized as a significant challenge in practice and in vocational education. The cultural adaptations of the NCCS made by Lin et al. [55] or Gözümet al. [56] also received a 4-factor structure.

**Table 6. Correlation of the NCCS-P scale with selected sociodemographic data.**

| Subscale | Age | | Job seniority | | Did you look after a patient from another culture? | | | | | | | |
|---|---|---|---|---|---|---|---|---|---|---|---|---|
| | | | | | Yes | | No | | I do not know | | Statistics | |
| | r | p | r | p | M | SD | M | SD | M | SD | H | p |
| NCCS-CA-P | **0.166**[*] | **0.010** | **0.237**[**] | **0.001** | 36.42 | 4.13 | 35.55 | 4.43 | 34.08 | 3.22 | **7.670** | **0.022** |
| NCCS-CK-P | -0.016 | 0.805 | -0.061 | 0.421 | 36.07 | 5.31 | 31.97 | 5.79 | 33.69 | 4.73 | **25.099** | **<0.001** |
| NCCS-CSe-P | 0.076 | 0.244 | 0.052 | 0.490 | 30.83 | 4.09 | 28.38 | 5.03 | 30.04 | 4.50 | **13.230** | **0.001** |
| NCCS-CS-P | -0.055 | 0.401 | -0.105 | 0.164 | 47.67 | 9.23 | 39.40 | 11.02 | 42.12 | 9.84 | **30.000** | **<0.001** |
| Total NCCS-P | 0.019 | 0.775 | -0.009 | 0.909 | 151.00 | 17.73 | 135.30 | 20.52 | 139.92 | 16.61 | **32.609** | **<0.001** |

NCCS-CA-P: Cultural Awareness Subscale; NCCS-CK-P: Cultural Knowledge Subscale; NCCS-CSe-P: Cultural Sensitivity Subscale; NCCS-CS-P: Cultural Skills Subscale; M- mean; SD-standard deviation r-Pearson correlation coefficient; H-test statistics for the Kruskal-Wallis test

[*]p<0.01

[**]p<0.001

The reliability of the Polish version of the NCCS-P was assessed by Cronbach's alpha at 0.95 for NCCS-CS-P; 0.72 for NCCS-CA-P; 0.77 for NCCS-CSe-P; 0.87 for NCCS-CK-P and 0.94 for the overall NCCS-P scale, which is considered more than acceptable in terms of consistent reliability [57] The Turkish Version of the Nurse Cultural Competence Scale (NCCS-T) achieved a Cronbach alpha of 0.96 in a trial of 235 nurses [56]. For the Taiwanese version of the NCCS scale, meanwhile, Cronbach's alpha coefficient of reliability was 0.88 for a trial of 246 nurses [55]. The Italian version of the NCCS verified in the center in Milan obtained satisfactory values of statistical analysis, i.e. consistency (alfa = 0.91), content validity (CVI = 95.8) and factor structure (confirmatory factor Index = 0.976, Tucker-Lewis Index = 0.987, RMSEA = 0.040, SRMR = 0.029). The alpha coefficient of the confirmatory factor solution was 0.90 [58].

The results for the reliability of the Polish NCCS-P scale are comparable to the original construct version authored by S. Perng, R. Watson, for a trial of 172 students the Cronbach's α coefficient for the four constructs ranged from 0.78 to 0.96 [51]. Work carried out on the development of the NCCS scale by an extensive team of Taiwanese researchers made it possible to retain four factors comparable to the original version as: aptitude for cultural awareness, aptitude for cultural action, aptitude for cultural sensitivity and aptitude for cultural knowledge; these were generated by exploratory factor analysis. These factors account for 62.0% of total variance [55].

## Level of cultural competence

Validation tests of the NCCS-P scale on the population of Polish nurses allow for an initial analysis of the cultural competences of the nurses surveyed. The cultural competence of the nurses, according to our study, takes on an average result (M = 143.92, SD = 20.07; out of a possible maximum 205), similar to a study of Polish, Czech, and Slovak Intensive Care Unit (ICU) nurses (most of those studied were of the Catholic religion) [24]. In the self-description of cultural competences, the Polish nurses surveyed indicated competences higher in the fields of cultural awareness and sensitivity than in skills and knowledge. Dobrowolska and her team have written about the low cultural skills of Polish and Czech nurses [24]. Similar results among American Associate Degree Nursing Students (most of whom were of Protestant denominations) indicating the highest results in cultural sensitivity and cultural awareness, moderate cultural skills, and the lowest results in cultural knowledge have been obtained by Hartman [59], as well as other authors [60]. Licentiate students at the University of Texas

achieved an average result of 162.3 (SD = 21.7) for cultural competence, with the highest value for cultural skills (M = 54.3; SD = 9.2), in a reverse tendency from our research. The American students achieved a rather high value for the cultural awareness category (M = 41.5; SD = 7.2) [61], which is consistent with our research results. The research showed nurses' awareness of the importance of taking cultural origin into account in providing high quality care, and of the importance of nursing education in the process of understanding the patient's cultural differences. Harkess and Kaddoura [62] have written on education as a factor influencing the formation of cultural competence, and above all in the area of cultural awareness, based on a review analysis of cultural competences, undergraduate programs and nursing teaching strategies.

Meanwhile, Lin and Hsu proved, in a randomized controlled trial, the effectiveness of the impact of an educational course on cultural competences on nurses' self-evaluation of their cultural competences [63]. Marzilli and Mastel-Smith [61] have also pointed to the necessity of culture education, preferably through contact with representatives of other cultures, which our study also confirms. Almutairi and his team [64] presented similar conclusions regarding nurses based on a study of a culturally diverse (23 countries) group of nurses. Other studies have also confirmed that cultural competence was significantly higher in groups which had educational experience in promoting cultural competence [65, 66] as well as in a group of students from a foreign exchange program [67], although the differences were smaller than might have been expected [24].

Our own research showed a high level of tolerance of the beliefs or behavior of various cultural groups in relation to health/illness and the ability to meet the needs of patients from different cultures by implementing nursing activities, increasing, as other authors indicate, the ability to act beyond cultural barriers [42, 68], which include country of residence, gender, age, year of studies, participation in cultural training, experience in caring for patients from other cultures or from special population environments, and living in a multicultural environment [10].

The openness to cultural differences highlighted in the opinion of the nurses surveyed suits the report demonstrating that nurses express a desire to help coworkers, patient caregivers, and even to help other patients in dealing with various cultural aspects in clinical settings [69]. The phenomenon of sensitivity to cultural difference is especially widely analyzed in Saudi Arabia, where foreign nurses make up approximately 66.4% of all nurses. Research results brought out in this multicultural population of nurses, speaking more than 40 languages, showed gaps in cultural competence and that healthcare leaders should be aware of the problems associated with this and be able to perceive and solve them [69, 70].

The cultural knowledge of the nurses studied, acquired through experience and observation of everyday lifestyle practices, referred to as procedural [knowledge] is of a higher level. Structural knowledge, on the other hand, regarding relationships between facts, procedures, and events, is somewhat lower, which most probably has something to do with the methods of shaping cultural competences. Possessing cultural knowledge generates an image of social reality; it initiates action in cultural competence, constituting a source of satisfaction and the basis for developing the personality of a culturally sensitive nurse [71, 72]. Italian nurses, among whom every second one had taken part in training in cultural diversity, presented moderate cultural competences, showing a moderately high level of cultural awareness and sensitivity (M = 5.41; SD = 0.66) and a moderate level of culturally competent behavior (M = 4.33; SD = 1.10). The growing cultural diversity of patients and their expectations concerning healthcare, however, is necessitating a systematic investment among nurses in developing cultural competence [73].

Cultural awareness competence in Polish nurses shows a positive correlation with age and seniority, which is not confirmed by other studies [24, 59, 61]. Instead, the majority of studies

show a relationship between nurses' experience in caring for patients of other cultures and the development of cultural competence.

## Limitations of the research project

There are some limitations to this study. The study used an intentionally moderately-sized sampling; therefore, no generalization of the results to the overall population of Polish nurses can be assumed. Consequently, further NCCS-P tests are necessary in order to cross-check other Polish clinical nurses. A second limitation is that over 32% of the variance was impossible to clarify because the NCCS-P is a new instrument with only preliminary tests, and there is still a need for additional tests to confirm the results of this study. The results of the Lin et al. review indicate that no one single instrument is suitable for assessing cultural competences in all contexts [55]. Another limitation is that, although several methods for determining the psychometric properties of the tool were assessed, the self-description format applied in the scale estimations does not leave an opportunity for excluding the possible impact of societal expectations. In future studies, societal expectations should be gauged for their impact on participants' responses.

## Implications for nursing

Care for culturally diverse clients is a neglected area of nursing education, research, and practice in traditionally homogeneous societies, including Poland. Cultural diversity is however a global phenomenon, and cultural competence is seen as a basic nursing skill. Despite the possible limitations of this study, the NCCS-P proves to be valid and reliable, and may be the first verified scale developed for the needs of a monocultural society transforming into a multicultural society.

The NCCS-P evaluation can help clinical managers and nursing teachers in selecting areas for intervention in ongoing professional development and improving performance in providing culturally competent care. Furthermore, the NCCS-P can assist in preparing content for intervention programs and in evaluating the effectiveness of such programs to improve the cultural competence of nurses. The study presents one of the first reports on the cultural competence level of Polish nurses and reinforces current literature, emphasizing the need for ongoing formation aimed at increasing cultural competence among nurses.

## Clinical significance

Providing culturally competent care is associated with better service provider-client communication, greater satisfaction with care, and improved health in the full understanding of health, compliance with recommendations on medication and lifestyle, and appropriate use of the healthcare system. Healthcare professionals must be adequately trained in order to provide culturally competent healthcare. This study presents for the first time a report on the topic of the level of cultural competence of Polish nurses and reinforces the current literature, emphasizing the need for ongoing formation with the goal of increasing cultural competence among nurses.

## Conclusions

1. The cultural competence of the nurses studied is of a moderate level. The nurses achieved higher values in the fields of cultural awareness and sensitivity, and lower values in the fields of cultural knowledge and skills.

2. The development of nurses' cultural competence is conditioned by care for patients of other cultures and cultural awareness coming with age and professional work experience.

3. The language adaptation shows that the Polish version of the NCCS-P questionnaire for assessing the cultural competence of nurses was understandable for a representative population and meets the psychometric criteria for reliability and accuracy of the tool.

## Supporting information

**S1 Checklist. STROBE statement—checklist of items that should be included in reports of cross-sectional study.**
(DOC)

**S1 Table. The model matrix<sup>a</sup> for Table 2.**
(DOCX)

## Acknowledgments

We would like to kindly thank Shoa-Jen Perng and Roger Watson for permission to Polish adaptation and validation of the Nurse Cultural Competence Scale. In addition, we would like to thank Katarzyna Szczekala and Wojciech Błachnio for their help in translating the first version of the NCCS-P.

## Author Contributions

**Conceptualization:** Agnieszka Chrzan-Rodak, Jadwiga Bąk, Barbara Ślusarska.

**Data curation:** Agnieszka Chrzan-Rodak, Jadwiga Bąk, Barbara Niedorys-Karczmarczyk.

**Formal analysis:** Danuta Zarzycka, Barbara Ślusarska.

**Funding acquisition:** Danuta Zarzycka, Barbara Ślusarska.

**Investigation:** Agnieszka Chrzan-Rodak, Jadwiga Bąk, Barbara Niedorys-Karczmarczyk, Barbara Ślusarska.

**Methodology:** Danuta Zarzycka, Agnieszka Chrzan-Rodak, Jadwiga Bąk.

**Project administration:** Danuta Zarzycka, Barbara Ślusarska.

**Resources:** Danuta Zarzycka.

**Software:** Barbara Ślusarska.

**Supervision:** Danuta Zarzycka.

**Validation:** Danuta Zarzycka.

**Visualization:** Danuta Zarzycka, Barbara Ślusarska.

**Writing – original draft:** Danuta Zarzycka, Agnieszka Chrzan-Rodak, Jadwiga Bąk, Barbara Niedorys-Karczmarczyk, Barbara Ślusarska.

**Writing – review & editing:** Danuta Zarzycka, Agnieszka Chrzan-Rodak, Jadwiga Bąk, Barbara Niedorys-Karczmarczyk, Barbara Ślusarska.

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
