## [Decision Letter · Decision Letter 0]

16 Jun 2020

PONE-D-20-07812

Nurse Cultural Competence-cultural adaptation and validation of the Polish version of the Nurse Cultural Competence Scale and preliminary research results

PLOS ONE

Dear Dr. Chrzan-Rodak,

Thank you for submitting your manuscript to PLOS ONE. After careful consideration, we feel that it has merit but does not fully meet PLOS ONE’s publication criteria as it currently stands. Therefore, we invite you to submit a revised version of the manuscript that addresses the points raised during the review process.

Minor revisions are required to your manuscript as detailed by both reviewers and we would request that you make these before resubmitting your paper. 

We look forward to receiving your revised manuscript.

Kind regards,

Fiona Cuthill, PhD

Academic Editor

PLOS ONE

Journal Requirements:

2. Please clarify in your Methods section whether the questionnaire is published under a CC-BY license, or whether you obtained permission from the publisher to reproduce the questionnaire in this manuscript. Please explain any copyright or restrictions on this questionnaire.

3. Please include your tables as part of your main manuscript and remove the individual files. Please note that supplementary tables (should remain/ be uploaded) as separate "supporting information" files.

4. Please include captions for your Supporting Information files (STROBE_checklist_Article_NCCS.doc changed from "other" to "supporting information" item type) at the end of your manuscript, and update any in-text citations to match accordingly. Please see our Supporting Information guidelines for more information: http://journals.plos.org/plosone/s/supporting-information.

Reviewers' comments:

Reviewer's Responses to Questions

**Comments to the Author**

1. Is the manuscript technically sound, and do the data support the conclusions?

Reviewer #1: Yes

Reviewer #2: Yes

2. Has the statistical analysis been performed appropriately and rigorously? 

Reviewer #1: Yes

Reviewer #2: Yes

3. Have the authors made all data underlying the findings in their manuscript fully available?

Reviewer #1: Yes

Reviewer #2: Yes

4. Is the manuscript presented in an intelligible fashion and written in standard English?

Reviewer #1: Yes

Reviewer #2: Yes

5. Review Comments to the Author

Reviewer #1: Thank you, this is a very good study and it is good to see confirmation of the structure and utility of this scale in another language and culture. The manuscript is very well written. I only have a few suggestions:

- the abstract should report the factor structure

- for the translation procedure it might be good to give a reference to a standard procedure eg the WHO procedure (If you used it: https://www.who.int/substance_abuse/research_tools/translation/en/) or whatever procedure and to say how closely you followed this.

- in Table 2 please present all the loadings on all the factors and not only those on putative factors

- minor points: Registered Nurse should have capitals; data are plural.

- finally in limitations maybe mention that an additional and now more conventional method estimating the number of factors to be rotated - Monte Carlo Parallel analysis - could have been used and, if you have not already said it then the structure could be studied using confirmatory factor analysis in future.

Reviewer #2: Thank you for the opportunity to review this article. Overall, this is an important topic and is well written. Below are just a few comments.

1. The author(s) use the term "cultural meetings" in reference to Campihna-Bacote's model and tool. However, the term is cultural encounter. Conceptually, meetings and encounter can be defined differently.

2. I would add some previous research findings to show validity and use of the tool.

3. There are some grammatical and flow issues within the article.

4. The statistics are sound and are congruent with other research findings in this area. I am not well-versed on how well studied this topic is in Poland. I am not sure what new findings this adds to the literature.

6. PLOS authors have the option to publish the peer review history of their article (what does this mean?). If published, this will include your full peer review and any attached files.

Reviewer #1: Yes: Roger Watson

Reviewer #2: No

---

## [Author Response · Author response to Decision Letter 0]

4 Sep 2020

Response 1:

Dear Reviewer,

Thank you very much for your opinion and all of your suggestions for making our manuscript better.

We added suggested extract in the abstract, page 1-2: The Cronbach's alpha reliability coefficient for the NCCS-P scale was 0.94, with the subscale values ranging from 0.72 to 0.95. Factor validity analysis of the Polish adaptation of the NCCS-P scale pointed to its four-factor structure. The Kaiser-Mayer-Olkin sampling adequacy test was 0.905, and the Bartlett test of sphericity result was χ2 = 5755.107; df = 820; p < 0.001. The four-factor structure is affirmed by the Kaiser criterion and the scree test result.

About translation procedure this content has been added on page 10: As part of the adaptation procedure of the original version, the NCCS was translated into Polish by two independent translators. Next, a panel of experts made up of nurses of varying professional experience and working in various positions, developed version 1.0 of the NCCS scale, which was re-translated into English by an independent translator who was a native speaker of English. After a comparison of the original and adapted 1.0 versions, a pilot study was conducted with 15 nurses. The interviewer, who was one of the authors of the document, held discussions with the pilot study participants on the clarity and comprehensibility of the survey items, which made it possible to form the final version of the NCCS, which was subjected to the further examinations presented in this article. Every stage of the language adaptation of the scale has been documented.

The model matrix for Table 2 has been included in the supplementary materials.

Thank you for your observations about static analysis. These corrections have been made. In this paper, Exploratory Factor Analysis was conducted. The structure of data could be studied using confirmatory factor analysis (CFA) in future analyze. CFA will allow for testing the number of factors (latent constructs) that best fit the model being examined.

Response 2:

Thank you very much for your opinion and all of your suggestions for making our manuscript better. The changes have been made and are highlighted in the manuscript text. The translator, who is a native speaker of English, has made a fresh revision of the manuscript.

The following current research results have been added:

-M-H, Lin, H-C Hsu, Effects of a cultural competence education programme on clinical nurses: A randomised controlled trial, Nurse Education Today,Volume 88, 2020, 104385, ISSN 0260-6917, https://doi.org/10.1016/j.nedt.2020.104385].

-Terzoni S, Ferrara P, Sala E, Destrebecq A, Trombetta N. La versioneitalianadella Nurse Cultural Competence Scale: validazionesuglistudenti di Infermieristica [The Italian version of the Nurse Cultural Competence Scale: validation on a sample of Nursing students]. Prof Inferm. 2020;73(1):13-20. doi:10.7429/pi.2020.721011

In the Polish literature there is a palpable lack of tools for assessing cultural competences of nurses as significant providers of health services and caregivers. The Polish version of the NCCS will make it possible to assess the current state of the cultural competences of nurses as well as to monitor their development in a reasonable way.

---

## [Decision Letter · Decision Letter 1]

6 Oct 2020

Nurse Cultural Competence-cultural adaptation and validation of the Polish version of the Nurse Cultural Competence Scale and preliminary research results

PONE-D-20-07812R1

Dear Dr. Chrzan-Rodak,

We’re pleased to inform you that your manuscript has been judged scientifically suitable for publication and will be formally accepted for publication once it meets all outstanding technical requirements.

Kind regards,

Fiona Cuthill, PhD

Academic Editor

PLOS ONE

Additional Editor Comments (optional):

Reviewers' comments:

Reviewer's Responses to Questions

**Comments to the Author**

1. If the authors have adequately addressed your comments raised in a previous round of review and you feel that this manuscript is now acceptable for publication, you may indicate that here to bypass the “Comments to the Author” section, enter your conflict of interest statement in the “Confidential to Editor” section, and submit your "Accept" recommendation.

Reviewer #1: All comments have been addressed

Reviewer #2: All comments have been addressed

2. Is the manuscript technically sound, and do the data support the conclusions?

Reviewer #1: Yes

Reviewer #2: Yes

3. Has the statistical analysis been performed appropriately and rigorously? 

Reviewer #1: Yes

Reviewer #2: Yes

4. Have the authors made all data underlying the findings in their manuscript fully available?

Reviewer #1: Yes

Reviewer #2: Yes

5. Is the manuscript presented in an intelligible fashion and written in standard English?

Reviewer #1: Yes

Reviewer #2: Yes

6. Review Comments to the Author

Reviewer #1: Thanks you for attending to all the comments that I made this is now a much clearer manuscript and presents the study much better.

Reviewer #2: Thank you for addressing the recommendations I made in the first submission. The authors addressed my concerns regarding the statistical analysis. They also added a current reference. Overall, the grammatical and flow issues were corrected.

7. PLOS authors have the option to publish the peer review history of their article (what does this mean?). If published, this will include your full peer review and any attached files.

Reviewer #1: **Yes: **Roger Watson

Reviewer #2: No

---

## [Editor Report · Acceptance letter]

8 Oct 2020

PONE-D-20-07812R1 

Nurse Cultural Competence-cultural adaptation and validation of the Polish version of the Nurse Cultural Competence Scale and preliminary research results 

Dear Dr. Chrzan-Rodak:

I'm pleased to inform you that your manuscript has been deemed suitable for publication in PLOS ONE. Congratulations! Your manuscript is now with our production department. 

Kind regards, 

on behalf of

Dr. Fiona Cuthill 

Academic Editor

PLOS ONE